# Evidence-guided approach to portfolio-guided teaching and assessing communications, ethics and professionalism for medical students and physicians: a systematic scoping review

Jacquelin Jia Qi Ting,[1,2] Gillian Li Gek Phua [iD],[2,3] Daniel Zhihao Hong,[1,2] Bertrand Kai Yang Lam,[2] Annabelle Jia Sing Lim,[1,4] Eleanor Jia Xin Chong,[1,2] Anushka Pisupati,[1,2] Rei Tan,[1,2] Jocelyn Yi Huang Yeo,[1,2] Yi Zhe Koh,[1,2] Chrystie Wan Ning Quek,[1,2] Jia Yin Lim,[1,2] Kuang Teck Tay,[2] Yun Ting Ong,[1,2] Min Chiam,[5] Jamie Xuelian Zhou,[2,3] Stephen Mason,[6] Limin Wijaya,[7] Lalit Kumar Radha Krishna [iD] [1,2,4,5,6,8]

For numbered affiliations see end of article.

**Correspondence to**
Dr Lalit Kumar Radha Krishna;
lalit.radha-krishna@liverpool.ac.uk

## ABSTRACT

**Objectives** Guiding the development of longitudinal competencies in communication, ethics and professionalism underlines the role of portfolios to capture and evaluate the multiple multisource appraisals and direct personalised support to clinicians. However, a common approach to these combined portfolios continues to elude medical practice. A systematic scoping review is proposed to map portfolio use in training and assessments of ethics, communication and professionalism competencies particularly in its inculcation of new values, beliefs and principles changes attitudes, thinking and practice while nurturing professional identity formation. It is posited that effective structuring of portfolios can promote self-directed learning, personalised assessment and appropriate support of professional identity formation.

**Design** Krishna's Systematic Evidence-Based Approach (SEBA) is employed to guide this systematic scoping review of portfolio use in communication, ethics and professionalism training and assessment.

**Data sources** PubMed, Embase, PsycINFO, ERIC, Scopus and Google Scholar databases.

**Eligibility criteria** Articles published between 1 January 2000 and 31 December 2020 were included.

**Data extraction and synthesis** The included articles are concurrently content and thematically analysed using the split approach. Overlapping categories and themes identified are combined using the jigsaw perspective. The themes/categories are compared with the summaries of the included articles in the funnelling process to ensure their accuracy. The domains identified form the framework for the discussion.

**Results** 12 300 abstracts were reviewed, 946 full-text articles were evaluated and 82 articles were analysed, and the four domains identified were indications, content, design, and strengths and limitations.

## STRENGTHS AND LIMITATIONS OF THIS STUDY

⇒ The Systematic Evidence-Based Approach methodology adopts the structure of systematic reviews and flexibility of narrative reviews to synthesise a reproducible and accountable evaluation of diverse methodological sources, settings, physician populations and specialities.

⇒ This review evaluates the impact of longitudinal development of communications, ethics and professionalism competencies and their impact on professional identity formation.

⇒ Given that communications, ethics and professionalism are sociocultural constructs, geopolitical sociocultural differences may raise questions as to the applicability of these findings beyond the European and North American medical education systems.

⇒ As the number of articles included is limited, the review's ability to assess the long-term effectiveness of portfolios may be compromised.

**Conclusions** This review reveals that when using a consistent framework, accepted endpoints and outcome measures, longitudinal multisource, multimodal assessment data fashions professional and personal development and enhances identity construction. Future studies into effective assessment tools and support mechanisms are required if portfolio use is to be maximised.

## INTRODUCTION

Evidence for the effective embodiment of ethical and professional principles, communication skills and appropriate use of empathy in clinical practice requires a

longitudinal and often multisource perspective. With ethics and professionalism sharing longitudinal developmental trajectories and intimately entwined with communication skills and competencies, the three competencies are increasingly considered together. In practice longitudinal communication, ethics and professionalism (CEP) programmes appear in the curricula of the top 10 medical schools featured on the QS World University Rankings 2020.[1] Concurrent study of CEP is also underlined by their common sociocultural roots. As sociocultural constructs,[2] CEP competencies are shaped by the individual's personal experiences, motivations, enthusiasm, idealism, abilities, competencies, virtues, expectations, knowledge, skills, emotions and attitudes (henceforth narratives); their values, beliefs and principles (henceforth belief systems)[3]; their clinical experiences, competencies, training, insights and confidence (henceforth clinical insights);[4–7] and their practice, clinical, social, cultural, academic, research and personal considerations (henceforth contextual considerations).[8–12]

However, while previous reviews into the teaching of ethics,[13–15] communication[16–22] and professionalism[23–25] suggest the use of portfolios could provide a personalised, holistic and longitudinal perspective of CEP skills, knowledge and attitudes and support of developing competencies, we are aware of little progress in designing such platforms. Impetus for mapping current use of CEP portfolios also arises from the notion that developing CEP competencies shapes how medical students and physicians (henceforth clinician) 'think, act and feel like a physician'[26] or their professional identity formation (henceforth PIF).[27] It is posited that the promise of CEP portfolios with better appreciation of evolving self-concepts of professional identity will better direct support and even remediation of professional, ethical, communication and interprofessional development and PIF. This is especially pertinent at a time of increasing reports of breaches in standards, codes of conduct, and social and practice expectations.[25]

## METHODS

A Systematic Evidence-Based Approach (SEBA) guided Systematic Scoping Review (SSR) (henceforth SSR in SEBA) is proposed to map CEP portfolio use in medicine to guide the design, structuring and support of a proposed programme.[13–17 20–22 24 25 27–50] Given space constraints, we briefly describe the six stages in the construction of SSRs in SEBA in figure 1 and more advanced details of the systematic approach, split approach, jigsaw perspective, funnelling process, reiterative process and synthesis of SSR in online supplemental appendix A.[16 27 28 35 41 48 51–56]

### Stage 1 of SEBA: systematic approach
#### Determining of title and background of the review
An expert team comprised of a medical librarian from the Yong Loo Lin School of Medicine (YLLSoM) at the National University of Singapore[57] and local education

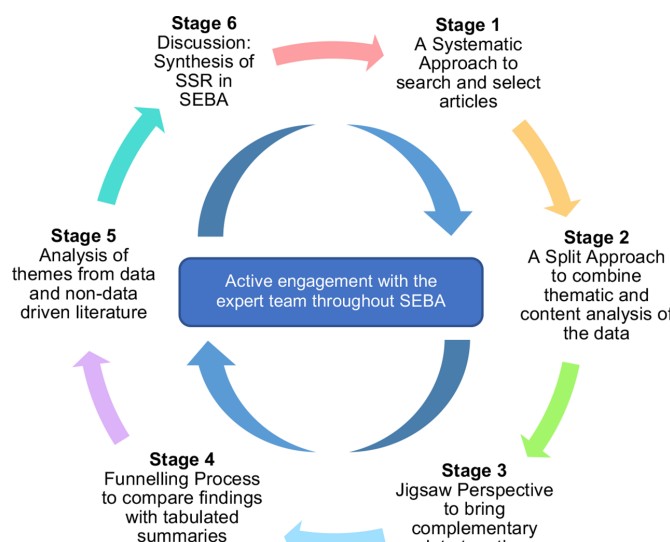

**Figure 1** The SEBA process. SEBA, Systematic Evidence-Based Approach.

experts and clinicians at the National Cancer Centre Singapore (NCCS), the Palliative Care Institute Liverpool, YLLSoM and Duke-NUS Medical School (henceforth the expert team) supported the research team in designing and overseeing the SEBA process.

### Identification of research question
The research and expert team determined the primary research question: '*What is known about CEP portfolios?*'. The secondary research questions were '*what role do CEP portfolios have in teaching and assessing CEP development?*'.

### Inclusion criteria
A population, intervention, comparison and outcome (PICOs) format, outlined in table 1, was used to guide the research process.[58 59]

### Identification of relevant studies
The research team developed search strategies and carried out independent reviews for relevant publications in the PubMed, Embase, PsycINFO, ERIC, Scopus and Google Scholar databases (search strategy enclosed in online supplemental appendix B). Keeping with Pham et al's[60] approach of ensuring a feasible and sustainable research process, the team contained the search to articles published between 1st January 2000 and 31st December 2020 to accommodate existing human resource and time constraints and ensure a sustainable review process.

### Selection of studies included in review
Six members of the research team created individual lists of titles to be included, while three other members of the research team carried out hand searches and ancestry searches of seven leading journals in medical education (*Academic Medicine, Medical Education, Medical Teacher, Advances Health Sciences Education, BMC Medical Education, Teaching and Learning in Medicine and Perspectives on Medical Education*) accessed through the National

**Table 1** PICOS inclusion and exclusion criteria

| PICOS | Inclusion criteria | Exclusion criteria |
|---|---|---|
| Population | ► Undergraduate and postgraduate medical students.<br>► Qualified medical doctors, physician or resident; medical officer, registrar, house officer, attending and consultant. | ► Allied health specialties such as pharmacy, dietetics, chiropractic, midwifery, podiatry, speech therapy, occupational and physiotherapy.<br>► Non-medical specialties such as clinical and translational science, alternative and traditional medicine, veterinary and dentistry. |
| Intervention | ► Portfolios in undergraduate and postgraduate medical education for teaching and assessment of communication, ethics and professionalism.<br>Criteria of a portfolio:<br>► Longitudinal (more than a single timepoint) assessment data.<br>► Candidate's personal engagement with portfolio content and associated learning.<br>► Interventions meeting the above criteria were included regardless of whether they were referred to as portfolios.<br>All types of portfolios were included in the study:<br>► For instance: electronic and non-electronic; formative and summative or combined; clinical and non-clinical.<br>► Portfolios with input from students and/or residents and/or doctors and/or input from faculty members and other individuals.<br>► Portfolios with different structures: extent by which the structure has been prescribed and/or left to individual discretion. | Other documentation methods or learning tools that are:<br>► Not longitudinal or single timepoint.<br>► Does not include personal intellectual engagement with the content and associated learning (for instance, curriculum vitae, logbooks and the use of personal digital assistants). |
| Comparison/context | NA | NA |
| Outcome | Papers that measured the following outcomes were included:<br>► Effectiveness of the use of portfolios to assess and teach communication, ethics and professionalism.<br>► Impact of the use of portfolios on medical students (both undergraduate and postgraduate).<br>► Impact of the use of portfolios on the faculty. | NA |
| Study design | ► Articles in English or translated to English.<br>► Articles published from 1st January 2000 to 31st December 2020.<br>► Databases: PsycINFO, Embase, PubMed, ERIC, Scopus and Google Scholar.<br>► All study designs including:<br>mixed methods research, meta-analyses, systematic reviews, randomised controlled trials, cohort studies, case–control studies, cross-sectional studies, descriptive papers, grey literature, opinions, letters, commentaries and editorials. | NA |

PICOS, population, intervention, comparison and outcome.

University of Singapore (NUS) library portal. These individual lists of titles were discussed online and Sandelowski and Barroso[61] 's 'negotiated consensual validation' approach to '*articulate, defend, and persuade others of the "cogency" or "incisiveness" of their points of view or show their willingness to abandon views that are no longer tenable*' was applied to achieve consensus on the final list of titles to be scrutinised.

### Assessing the quality of articles

Eight research team members individually appraised the quality of the quantitative and qualitative studies using the Medical Education Research Study Quality Instrument[62] and Consolidated Criteria for Reporting Qualitative Studies.[63]

### Stage 2 of SEBA: split approach

Two teams carried independent and concurrent thematic and content analysis of the included articles while a third team created tabulated summaries based on recommendations drawn from Wong *et al's*[64] RAMESES publication standards: meta-narrative reviews, and Popay *et al's*[65] '*Guidance on the conduct of narrative synthesis in systematic reviews*'. The categories employed in the content analysis for undergraduate communications were Rider *et al's*[66] '*A model for communication skills assessment across the undergraduate curriculum*', Goldie's[67] '*Review of ethics curricula in undergraduate medical education*', Duffy *et al's*[68] '*Assessing Competence in Communication and Interpersonal Skills: The Kalamazoo II Report*' and Hong *et al's*[13] '*Postgraduate Ethics Training Programs: A Systematic Scoping Review*'. Tay *et al's*[24] '*Assessing Professionalism in Medicine - A Scoping Review of Assessment Tools from 1990 to 2018*' was employed for codes for professionalism, and Friedman Ben David *et al's*[69] article '*AMEE Medical Education Guide No. 24: Portfolios as a method of student assessment*' was then used to contextualise their use in portfolios.

### Stage 3 of SEBA: the jigsaw perspective

The jigsaw perspective sees the themes and categories identified compared and combined where overlaps and similarities exist.

### Stage 4 of SEBA: the funnelling process

The funnelling process sees the themes/categories created from the jigsaw approach compared with the tabulated summaries to determine their consistency.

### Patient and public involvement

Patients or the public were not involved in the design, or conduct, or reporting, or dissemination plans of our research.

### RESULTS

A total of 12 300 abstracts were reviewed, 946 full-text articles were evaluated and 82 articles were analysed (figure 2). The funnelled domains identified are: (1) indications, (2) portfolio content, (3) portfolio design

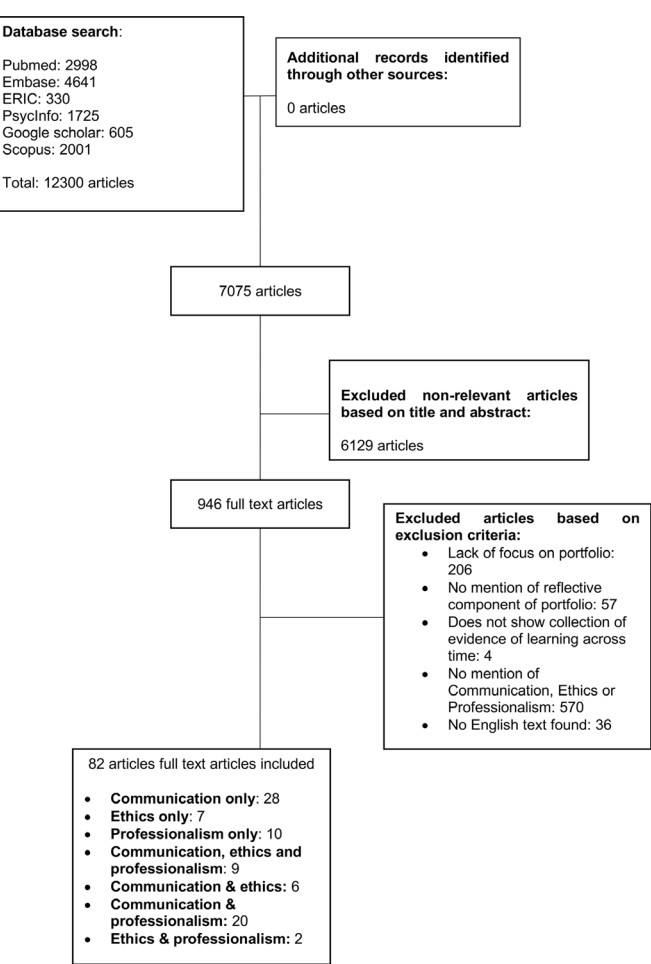

**Figure 2** PRISMA flow chart. PRISMA, Preferred Reporting Items for Systematic Reviews and Meta-Analyses.

and implementation and (4) strengths and limitations. In keeping with its goals of transparency and accountability, these tables are enclosed in online supplemental appendix C.

### Funneled domain 1: indications

CEP portfolios curate longitudinal multisource, multimodal assessment data taken at different settings and contexts allowing it to serve two main roles: teaching and assessment[70 71] (table 2). However, few span undergraduate and postgraduate education.[69 72 73] As assessment tool CEP portfolios assist in the identification of areas of weakness and guide the provision of a comprehensive, targeted feedback, support and remediation.[72 74 75] Most CEP assessment portfolios in postgraduate education focus on evidencing the attainment of required competencies,[76–80] capturing reflections and indications of PIF.[78 81–83] CEP portfolios in the postgraduate setting have also been used for revalidation purposes.[81]

As a teaching tool, CEP portfolios encourage self-directed learning,[75 84–86] self-improvements,[71] reflective practice,[72 73 75 79 81 85–95] motivate practitioners to achieve desired competencies[90 96] and bolster overall professional and personal development.[72 88 96 97]

**Table 2** Benefits of CEP teaching and assessment portfolios

| CEP teaching portfolios | CEP assessment portfolios |
|---|---|
| **Enhancing learning**<br>Promotion of life-long learning[104 106]<br>Promotion of self-directed learning<br>► Formulation of learning plans.[90 96 126]<br>► Facilitates outcome-oriented learning.[89 90 118 120]<br>► Enhances self-evaluative capabilities.[73 76 80 85 88 89 100 104 106 112 126 127]<br>► Identification of strengths and weaknesses.[69 71 76 78 92 96 100 101 109 128 129]<br>► Inculcates responsibility and ownership for one's learning.[71 73 74 80 84 86 113 118]<br>► Sets the pace.[104]<br>► Personalised learning.[109 129]<br><br>**Personal and professional development**<br>Promotes self-understanding[79 114 115]<br>► Understanding feelings and thoughts.[126 129]<br>► Understanding of one's values.[92]<br>► Understanding activity/experience.[69 88 123]<br>► Understanding of how one functions in a team.[122]<br>► Fears and stressors feelings of inadequacy.[123 137]<br>Curates evidence of skills[106]<br>► Skills to compile a portfolio.[102 106 130 138 139]<br>► Reflective skills.[69 76 88 120 123 129 140]<br>► Clinical skills.[104]<br>Organises information[120 128]<br>► Documentation of progress.[77 100 101 106]<br>► Documentation of skills in different settings.[71 101 109]<br>Cross-sectional view of one's competence[69 77 101 109]<br><br>**Personalisation**<br>Allows for personalisation[69 90 104 113 132]<br>► Personal experiences.[69]<br>Choice over the contents[69 104 132]<br><br>**Benefits to faculty**<br>Improvement of curriculum[83 92]<br>► Identify areas of weakness in curriculum.[83]<br>► Provide insights into mentoring, teaching, etc.[141]<br> Facilitates design of the programme.[83]<br> Faculty development<br>► Understand more about how students acquire competencies.[104]<br>► Insight into the impact of their interventions.[136 141]<br>Insight into their professional conduct, role modelling and pastoral care[115]<br><br>**E-portfolios**<br>Ease of use:<br>► Easier use.[90]<br>► Accessible.[73]<br>Dissemination and editing of information[113] | Balance of structure and freeform learning[69 76 92 128 130 131]<br>Adaptable to goals[77 98 132]<br>Alignment of learning outcomes and assessment methods[90 91 97 112 118]<br>Allows personalised assessments[77]<br>Allows assessment over time[84 92] and stages of development[69] and that facilitate evaluation of:<br>► Learning and progress over time.[77 101]<br>► Learning curve.[115]<br>► In different roles.[92 93]<br>► In different settings.[70 98]<br>► Complex behaviours.[72 74 77 84 90 104 122 133]<br>► Critical thinking.[77 97 112 132]<br>► Outcomes/competencies.[128 133]<br>► Self-assessment.[77 112 132]<br>► Complex skills.[98]<br>► Professionalism.[132]<br>► Cultural competence.[132]<br>► Communication skills.[132]<br>► Clinical ethics.[134]<br>► Problem solving abilities.[77 132]<br>► Reflective abilities.[77 90 109 112]<br>Longitudinal data<br>► Possess good predictive value of subsequent clinical competency.[74 135]<br>► Enhances validity of portfolio grading.[69]<br>► Guides better decisions.[74 122 136]<br>► Provides authentic assessment is a key principle of it.[92 104 109]<br>► Provides insights in the various core competencies.[106]<br>► Fair mode of assessment[104] and allows student to defend positions/work.[69]<br>► Provides students with direction and clarity.[130 131]<br>Allows the combination of summative and formative assessment[69 91] and multisource assessments[81 87 97 129 109] and triangulation[77 109]<br>Facilitates personalised feedback[122] that can be informed by understanding of the individual setting and context[98 104 126] |

CEP, communication, ethics and professionalism.

## Funneled domain 2: characteristics

The contents of portfolios were shaped by their overall function[77 80 83 98 99] (table 3). Expectations about the nature and assessments of the contents[83] facilitate personalisation of portfolios.[73 77 100]

Postgraduate CEP assessment portfolios inject more details on undergraduate portfolios. Table 4 summarises these elements include communication skills, addressing ethical issues and compliance of professional, institutional and departmental policies.[101–103] Greater detail in postgraduate CEP assessment portfolios sees the addition of modified essay questions,[79] polypharmacy journal/audits,[70 95 99 104 105] professional workshops and training sessions,[98 104 106–108]

**Table 3** Portfolio design and implementation

| Themes | Subthemes | Contents |
|---|---|---|
| Design and implementation principles | Structure | Clear guidelines[69 87 88 96 97 112 129]<br>▶ Use of portfolio.[69 88]<br>▶ Portfolio content – determining competencies to assess.[69 91]<br>▶ Promotional decisions.[69 71 89 115 142]<br>Standard-setting exercises[75]<br>Provision of sample portfolios for reference[112 116 121]<br>▶ Roles for assessors.[69 71 74 114 115 117 142]<br>▶ Expectations of students.[131]<br>▶ Duration spanning across preclinical and clinical years.[139]<br>Well-defined structure with room for flexibility[69 86 88 90 112]<br>Student-centred[90]<br>Appropriate student to mentor ratio[71 138]<br>Sufficient curricular experience for reflection and feedback[71]<br>Protected time for reflection, feedback and regular meetings with mentors[71 74 87 129] |
| | Faculty | Good faculty support.[115 139]<br>Ensure culture is supportive of portfolio system.[71 139]<br>Flexible and enthusiastic tutors passionate and experienced mentors.[85 88] |
| | Curricular integration | Early introduction of portfolio[88 113 129 143]<br>Easing portfolio into curriculum[69 72 86 88 90] – including orientation, training and introductory sessions[69 72 88 90 112 115 116] for learners and staff training[69 71 72 75 84 85 88]<br>Implementation of portfolios throughout curriculum[115]<br>Portfolio content aligning with phase of medical journey and competencies required[69 71 90 97 116 142] |
| | Assessments | Clear scoring rubrics[69 139]<br>Alignment between assessments and outcomes[86 87 90]<br>Transparent assessment criteria and outcomes to maintain fairness[71 72 113 121]<br>Standardisation of portfolio use and content,[86 87] assessors training[72] and increased number of assessors[86 87 143] to improve assessment reliability[86 87 143]<br>Separate processes and reviewers for summative and formative portfolio[71] – summative assessments to specify clear pass/fail requirements[69]<br>Incorporation of student feedback into portfolio development[69 72 90] |
| | Assessors | External examiners[69]<br>Faculty members[69 71 74 114 117 142]<br>Mentors[128]<br>Senior staff to be paired with new examiners[69]<br>Mentors should not assess own student[86 92] |
| | Assessment frequency | Formative assessments to monitor and guide student learning – conducted throughout the year[86 88 91]<br>Summative assessments to evaluate learner performance – conducted annually,[69 71 72 74 75 84 85 90 91 116 129 142 144–147] twice,[71 75 122] thrice[88] or multiple times a year[77 79] |
| | Oversight and delegation | Establishment of portfolio oversight committee[71 75]<br>Regular programme evaluation[71]<br>Clear division of support roles – for example, pedagogical and technical support[113] |

health service meeting reports[79] and reviews by medical regulatory authorities.[95] A detailed summary is provided in online supplemental appendix D.

### Funnelled domain 3: strengths and limitations

The strengths and limitations of portfolios are outlined in table 5.

Postgraduate portfolios provide greater insights into lifelong learning and continuous professional development.[77 80 81 83 98 101 104 106 107 109] Use of multi-source assessments at multiple timepoints[80] in clinical settings[81 104] see postgraduate portfolios provide good inter-rater reliability.[77 109] Postgraduate portfolios also detect stress and burnout.[107]

### Stage 5 of SEBA: analysis of data and non-data driven literature

Findings from the different stages of SEBA were discussed with members of the expert team and relevant stakeholders. There were concerns from the expert team about the impact of grey literature on the narrative, given that they were neither peer reviewed nor necessarily evidence based. As a result, the research team differentiated correspondence, letters, editorials and perspective pieces from

**Table 4** Portfolio content (competencies assessed and assessment modalities)

| | Communication | Ethics | Professionalism |
|---|---|---|---|
| Common competencies assessed | 1. Values/ethics – appreciation of professional, legal and societal values to guide practice.<br>2. Professional responsibilities – appreciation and manifesting these responsibilities.<br>3. Doctor–patient relationships- appreciation of the guiding principles, expectations and ability to nurture these relationships.<br>4. Interprofessional team working.<br>5. Ethical and legal responsibilities.<br>6. Continuous learning and quality improvement. | | |
| Specific competencies | 1. Active listening.<br>2. Clear explanations.<br>3. Empathetic communication.<br>4. History taking.<br>5. Breaking bad news.<br>6. Conflict management.<br>7. Shared decision making.<br>8. Written communication. | 1. Professionalism.<br>2. Respect.<br>3. Privacy.<br>4. Identifying key ethical issues.<br>5. Balancing competing ethical responsibilities.<br>6. Consent. | 1. Interpersonal communication. |
| Reflection | 1. Content.<br>2. Context.<br>3. Depth.<br>4. Authenticity.<br>5. Developmental areas.<br>6. Dilemma and doubts. | | |
| Assessment (formative/summative/ mixed) | MCQ<br>Essays<br>OSCE<br>Mini-CEX<br>Case presentations<br>Case based discussion<br>Simulated patient assessment<br>Work-based assessment<br>Team-based assessment<br>Peer evaluation<br>Self-evaluation<br>Faculty evaluations<br>Feedback<br>Reflections<br>Letters of commendations<br>Learning and or improvement plans<br>Curriculum vitae<br>Logbook review/interview | | |

academic databases and grey literature from data-driven or research-based peer-reviewed data. Both groups were then independently analysed. The themes/categories identified were then compared with enhance further the accountability and the reproducibility of stage 5 of SEBA. Evidence-based data from bibliographic databases (henceforth evidence-based publications) were separated from grey literature, perspectives, editorials, letters and non-data-based articles drawn from bibliographic databases (henceforth non-data driven). These two groups were separately thematically analysed.

### Stage 6 of SEBA: synthesis of the discussion

The Best Evidence Medical Education Collaboration Guide[110] and the Structured approach to the Reporting In healthcare education of Evidence Synthesis[111] were used to guide the discussion.

### DISCUSSION

This SSR in SEBA suggest that overlaps in ethics, professionalism and communication skills, knowledge, attitudes and competencies reaffirm the concurrent training and assessments of CEP competencies in portfolios.[72 74 75] To be effective CEP teaching and longitudinal assessment portfolios require a consistent framework replete with clearly delineated goals,[112] aligned expectations, predetermined assessment criteria for specified competencies,[69 71 72 77 83 85 87 96 97 102 103 106 112–117] agreed on endpoints and outcome measures and the curation, structuring and organisation of longitudinal multisource, multimodal assessment data.[70 71] The combination of these data facilitate training, assessment, feedback and holistic and longitudinal support,[75 84–86 101–103 118 119] guide reflection[72 73 75 79 81 85–95] and bolster professional and personal

**Table 5** Strengths and limitations

| Themes | Subthemes | Content |
|---|---|---|
| Strengths/ limitations | Strengths | For students<br>Identification of own strengths and weaknesses.[69 74 75 85–88 90 92 96 126 147]<br>Appreciation of effective communication,[88 143 148] ethical issues[92] and fosters critical professional attitudes.[87 149]<br>Encourages self-regulation[147] through opportunities for reflection[86] and self-assessment.[88 115 129]<br><br>Facilitates preparation for postgraduate medical training.[88 128 130]<br>For faculty<br>Identification of areas of concern and allows for early, timely intervention.[74 75]<br>Enhances interaction between students and teachers.[69 88 92]<br>Facilitates reiterative reassessment of teaching strategies.[69]<br>Assessments<br>Longitudinal[122]<br>Good validity.[74]<br>Reliable assessment tool – allows for triangulation of information and evidence from multiple sources and contexts.[71]<br>Ability to assess a wide range of competencies.[90 97]<br>Allows for more authentic assessments.[74]<br>Gives faculty greater insight into students' achievement of behaviourally oriented competencies that traditional methods do not.[75]<br>Evaluates achievement during clinical attachment/clerkship.[97]<br>Provides a basis for the judgement of a student's professional fitness to practice.[72] |
| | Limitations | Time consuming and labour intensive.[72 75 85 88–91 96 128 139 143]<br>Issues with reliability.[86 90 114 143]<br>Inconsistent effects.[86 90 129 150]<br>Lack of honesty by students in reflections.[71 72 86 90 91 128 129 131] |

development.[72 88 96 97] Critically, the employ of consistent and agreed on structure, goals, purpose, role, learning objectives, content and assessment guidelines also affords users with a personalised perspective of their CEP development on which they can reflect on and glean insights and feedback.[86 120 121]

Yet to be effective, CEP portfolios require policing and underscore the need for trained[70] and dedicated faculty[88] capable of effective role modelling,[115] support[69 79 88 92 97] and providing feedback[122] and who appreciate the individual's current concepts, narratives, motivations, abilities and availabilities and contextual considerations. Furthermore, the efficacy of portfolios and their ability to meet the stated objectives stated in table 2 and realise their stated benefits in table 5 are reliant on the appropriate mix of assessment methods employed, the frequency of assessments, the efficacy of analysis and the quality of the feedback, support and remediation provided should it be required. This in turn highlights the role of the host organisation in ensuring effective oversight of the programme and assessments and in ensuring that faculty are afforded protected time for administrative duties, coordination and assessments particularly given the longitudinal nature of CEP portfolios.[77 90 101 105 112 123–125]

It should not be forgotten that CEP portfolios reflect the development of an individual and such changes should be measured and contextualised within the wider education, training, practice, professional, research, clinical, interprofessional team based and organisational perspective.

This further underlines the importance of the host organisation's role in assessing and guiding development of CEP competencies.

With these requisites met, evidence of changes in CEP knowledge, skills and competencies suggests a shift in the thinking, attitudes and conduct of clinicians. The Krishna-Pisupati model posits that sustained changes in practice, beliefs, values and principles will result in changes in the clinician's belief systems which, in turn, inform their self-concepts of identity and shape their PIF.

## LIMITATIONS

The generalisability of the results of this study is limited by the identified studies selected for review. First, the majority of the papers included originate from Europe and North America. Given that CEP is a sociocultural construct, these geopolitical sociocultural differences raise questions as to the applicability of these findings beyond the European and North American medical education systems. Second, as the number of articles included is limited, the SSR in SEBA's ability to assess the long-term effectiveness of portfolios may be compromised.

## CONCLUSIONS

The suggestion that CEP portfolios can capture, instruct and assess PIF requires further study. While there have been posits on how such evaluations could be made, such

tools remain unrealised and must be the focus of coming studies. One possible starting point for the design of such tools may be theories such as the Krishna-Pisupati model, which attempts to link PIF to changes to belief systems and concepts of personhood. This may provide a grander view of PIF that spans the undergraduate and postgraduate settings and provide the basis for directing support and feedback for clinicians. As we look forward to furthering efforts to study the effects of CEP portfolios on the PIF of clinicians, we also believe the specific impact of these potentially resource-heavy and financially costly intervention should be at the centre of future initiatives.

**Author affiliations**
[1]Yong Loo Lin School of Medicine, National University of Singapore, Singapore
[2]Division of Supportive and Palliative Care, National Cancer Centre Singapore, Singapore
[3]Lien Centre for Palliative Care, Duke-NUS Medical School, Singapore
[4]Department of Supportive and Palliative Care, National Cancer Centre Singapore, Singapore
[5]Division of Cancer Education, National Cancer Centre Singapore, Singapore
[6]Palliative Care Institute Liverpool, University of Liverpool, Liverpool, UK
[7]Department of Infectious Diseases, Singapore General Hospital, Singapore
[8]Duke-NUS Medical School, Singapore

**Acknowledgements** The authors would like to dedicate this paper to the late Dr S Radha Krishna and Assistant Professor Cynthia Goh, whose advice and ideas were integral to the success of this review and to Thondy, Maia Olivia and Raja Kamarul Ariffin, whose lives continue to inspire us. The authors would also like to thank the anonymous reviewers, Dr Ruaraidh Hill and Dr Stephen Mason for their helpful comments that greatly enhanced this manuscript.

**Contributors** JJQT, GLGP, DZH, BKYL, AJSL, EJXC, AP, RT, JYHY, YZK, CWNQ, JYL, KTT, YTO, MC, XJZ, SM, LW and LKRK were involved in data curation, formal analysis, investigation, preparing the original draft of the manuscript as well as reviewing and editing the manuscript. LKRK is the guarantor. All authors have read and approved the manuscript.

**Funding** The authors have not declared a specific grant for this research from any funding agency in the public, commercial or not-for-profit sectors.

**Competing interests** None declared.

**Patient and public involvement** Patients and/or the public were not involved in the design, or conduct, or reporting, or dissemination plans of this research.

**Patient consent for publication** Not applicable.

**Ethics approval** Not applicable.

**Provenance and peer review** Not commissioned; externally peer reviewed.

**Data availability statement** No data are available.

**ORCID iDs**
Gillian Li Gek Phua http://orcid.org/0000-0002-2034-9723
Lalit Kumar Radha Krishna http://orcid.org/0000-0002-7350-8644

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
