## [Reviewer comments · BMJ Open]

ARTICLE DETAILS

TITLE (PROVISIONAL)	An evidence-guided approach to Portfolio guided Teaching and Assessing Communications, Ethics and Professionalism for medical students and physicians:A Systematic Scoping Review
AUTHORS	Ting, Jacquelin Jia Qi; Phua, Gillian Li Gek; Hong, Daniel Zhihao; Lam, Bertrand Kai Yang; Lim, Annabelle Jia Sing; Chong, Eleanor Jia Xin; Pisupati, Anushka; Tan, Rei; Yeo, Jocelyn Yi Huang; Koh, Yi Zhe; Quek, Chrystie; Lim, Jia Yin; Tay, Kuang Teck; Ong, Yun Ting; Chiam, Min; Zhou, Xuelian Jamie; Mason, Stephen; Wijaya, Limin; Krishna, Lalit

VERSION 1 – REVIEW

REVIEWER	Chen, Yan University of Auckland, Centre for Medical and Health Sciences Education
REVIEW RETURNED	11-Oct-2022

GENERAL COMMENTS	Thanks for the opportunity to review. I have no doubt that a lot of time and effort have go into the present review, and the authors did a tremendous job at describing the study procedure and synthesizing the results. The review is content heavy, and it is quite difficult to navigate its content and reach its conclusions in the paper's current format. It would benefit from having a more focused introduction to describe the key issues the review aims to address. For example, what do we already know about CEP portfolio and its impact on professional identity formation? What more do we need to know? Why does it matter? The authors need to explicitly state the research objectives of the systematic scoping review and justify their decision to conduct a SSR over other types of literature review. References are missing for the SSR in SEBA method. More justification is required to explain the publication time imposed (i.e., 1/01/2000 to 31/12/2020). It is crucial for the authors to include their search algorithm in the manuscript. The decision to combine the literature on the use of portfolios in medical students and practicing physicians needs further justification. It is unclear what the key findings of Stage 5 of SEBA are. The authors mentioned the Ring Theory of Personhood (RToP) as one of the strengths of the study. However, the RToP is not discussed at all in the main text of the paper.
---

REVIEWER	Khabaz Mafinejad, Mahboobeh Tehran University of Medical Sciences, Department of Medical Education
REVIEW RETURNED	20-Oct-2022

GENERAL COMMENTS	I am grateful for the opportunity to review this manuscript, which covers an interesting in the timely topic in medical education. I believe that the authors make a good case for the need for such a study. Their approach appears to be reasonable and well-documented. Although the following comments are meant to help improve this work.  [ ] Why did you search the literature only for the time period 2000 to 2020? Please provide a rationale for this search strategy. [ ] It is not acceptable to use abbreviations in the abstract section. Please revise it. [ ] The authors did not specify how to access journals to be hand-searched. [ ] You describe the strength of the study being the study itself, i.e., you state that a study strength is that you use SEBA and Ring Theory of Personhood. This is not a strength, it is simply what you found out from doing the study by this method. [ ] Please provide a more detailed explanation of the kappa coefficient between researchers who participated in 1) thematic analysis and 2) content analysis separately. [ ] The discussion section is long and seems to ramble a bit. The entire section needs to be better organized and clear. Perhaps you could add section headings and discuss key findings within each section. As is, it is very difficult to follow. [ ] The manuscript was well written. However, I found the methods and results section of the abstract difficult to understand. [ ] As the authors have mentioned: "Disagreements were solved by discussion and consensus was reached for different parts of data analysis. It is not clear how several researchers coded and how they reached to consensus. How many sessions/hours were used to reach a consensus?" Again, thank you for a progressive and relevant research study. This will lay important groundwork for more sophisticated teaching and assessing CEP in all disciplines. Well done!
--

VERSION 1 – AUTHOR RESPONSE

Reviewer: 1

Dr. Yan Chen, University of Auckland

Comments to the Author:

Thanks for the opportunity to review. I have no doubt that a lot of time and effort have go into the present review, and the authors did a tremendous job at describing the study procedure and synthesizing the results. The review is content heavy, and it is quite difficult to navigate its content and reach its conclusions in the paper's current format. Thank you and we hope we have addressed your comments

It would benefit from having a more focused introduction to describe the key issues the review aims to address.

For example, what do we already know about CEP portfolio and its impact on professional identity formation? What more do we need to know? Why does it matter?

The authors need to explicitly state the research objectives of the systematic scoping review and

justify their decision to conduct a SSR over other types of literature review. Thank you, we have rewritten the introduction to ensure a more focused approach

References are missing for the SSR in SEBA method. Thank you. We have added these references
More justification is required to explain the publication time imposed (i.e., 1/01/2000 to 31/12/2020).
Thank you. This is in keeping with Pham's approach and recognition of the time and resource limitations

It is crucial for the authors to include their search algorithm in the manuscript Thank you we will include this in the appendix

The decision to combine the literature on the use of portfolios in medical students and practicing physicians needs further justification. Thank you we have explained that postgraduate portfolios build on and deepen the data collected from undergraduate portfolios

It is unclear what the key findings of Stage 5 of SEBA are. Thank you. We have clarified this
The authors mentioned the Ring Theory of Personhood (RToP) as one of the strengths of the study. However, the RToP is not discussed at all in the main text of the paper. Thank you. We have removed this

Prof. Mahboobeh Khabaz Mafinejad, Tehran University of Medical Sciences

Comments to the Author:

I am grateful for the opportunity to review this manuscript, which covers an interesting in the timely topic in medical education. I believe that the authors make a good case for the need for such a study. Their approach appears to be reasonable and well-documented. Although the following comments are meant to help improve this work. Thank you for structured comments and we hope we have addressed them

→ Why did you search the literature only for the time period 2000 to 2020? Please provide a rationale for this search strategy. Thank you. This is in keeping with Pham's approach and recognition of the time and resource limitations

→ It is not acceptable to use abbreviations in the abstract section. Please revise it.

Thank you. We have edited this

→ The authors did not specify how to access journals to be hand-searched. Thank you we have clarified this facet

→ You describe the strength of the study being the study itself, i.e., you state that a study strength is that you use SEBA and Ring Theory of Personhood. This is not a strength, it is simply what you found out from doing the study by this method. Thank you. We have removed all mention of the RToP .

→ Please provide a more detailed explanation of the kappa coefficient between researchers who participated in 1) thematic analysis and 2) content analysis separately. Thank you. We have included a more detailed account of the SEBA methodology in Appendix 1 and explained the rationale for not evaluating the kappa coefficient

The discussion section is long and seems to ramble a bit. The entire section needs to be better organized and clear. Perhaps you could add section headings and discuss key findings within each section. As is, it is very difficult to follow. We do apologise. We have shortened it significantly

→ The manuscript was well written. However, I found the methods and results section of the abstract difficult to understand. Thank you we have rewritten this

→ As the authors have mentioned: "Disagreements were solved by discussion and consensus was reached for different parts of data analysis. It is not clear how several researchers coded and how they reached to consensus. How many sessions/hours were used to reach a consensus? Thank you we have clarified this in Appendix 1

VERSION 2 – REVIEW

REVIEWER	Khabaz Mafinejad, Mahboobeh Tehran University of Medical Sciences, Department of Medical Education
REVIEW RETURNED	23-Feb-2023

GENERAL COMMENTS	Thank you for revising the manuscript of ‘an evidence-guided approach to Portfolio guided Teaching and Assessing Communications, Ethics, and Professionalism for medical students and physicians: A Systematic Scoping Review’. It is obvious that the authors have carried out the proposed revisions and have responded to each of the comments. Many Thanks to give me a great opportunity to read this interesting article.  • I would suggest that the results of table 4 be summarized in the form of frequency to create a better view of the most frequent items included in the portfolio for the audience.
--

VERSION 2 – AUTHOR RESPONSE

Formatting Amendments (where applicable): Reviewer: 2 Prof. Mahboobeh Khabaz Mafinejad, Tehran University of Medical Sciences Comments to the Author: Thank you for revising the manuscript of ‘an evidence-guided approach to Portfolio guided Teaching and Assessing Communications, Ethics, and Professionalism for medical students and physicians: A Systematic Scoping Review’. It is obvious that the authors have carried out the proposed revisions and have responded to each of the comments. Many Thanks to give me a great opportunity to read this interesting article.  • I would suggest that the results of table 4 be summarized in the form of frequency to create a better view of the most frequent items included in the portfolio for the audience. 	Thank you. We have summarised Table 4 in the text and provided the detailed summary in Appendix C
--	--